# TEMPORAL GRAPH MODELS FAIL TO CAPTURE GLOBAL TEMPORAL DYNAMICS

## ABSTRACT

A recently released Temporal Graph Benchmark is analyzed in the context of Dynamic Link Property Prediction. We outline our observations and propose a trivial optimization-free baseline of "recently popular nodes" outperforming other methods on medium and large-size datasets in the Temporal Graph Benchmark. We propose two measures based on Wasserstein distance which can quantify the strength of short-term and long-term global dynamics of datasets. By analyzing our unexpectedly strong baseline, we show how standard negative sampling evaluation can be unsuitable for datasets with strong temporal dynamics. We also show how simple negative-sampling can lead to model degeneration during training, resulting in impossible to rank, fully saturated predictions of temporal graph networks. We propose improved negative sampling schemes for both training and evaluation and prove their usefulness. We conduct a comparison with a model trained non-contrastively without negative sampling. Our results provide a challenging baseline and indicate that temporal graph network architectures need deep rethinking for usage in problems with significant global dynamics, such as social media, cryptocurrency markets or e-commerce. We open-source the code for baselines, measures and proposed negative sampling schemes.

## 1 INTRODUCTION AND RELATED WORK

**Temporal Graphs (TGs)** are ubiquitous in data generated by social networks, e-commerce stores, video streaming platforms, financial activities and other digital behaviors. They are an extension of *static* graphs to a dynamic temporal landscape, making it possible to capture evolution of graphs. A number of machine learning methods on TGs have been developed recently (Rossi et al., 2020), (Trivedi et al., 2019), (Wang et al., 2022), (Wang et al., 2021), (Cong et al., 2023). However, their reliable benchmarking is still on open issue. Poursafaei et al. (2022) discovered that TG benchmarking methods do not reliably extrapolate to real-world scenarios. Huang et al. (2023) identified further problems: small size of datasets, inflated performance estimations due to insufficient metrics.

**Temporal Graph Benchmark (TGB)** by Huang et al. (2023) is a collection of challenging and diverse benchmark datasets for realistic evaluation of machine learning models on TGs along with a well designed evaluation methodology. It incorporates datasets with orders of magnitude more nodes, edges and temporal steps compared to previously available ones. We build upon the Temporal Graph Benchmark to further improve TG model benchmarking methods.

**Dynamic Link Property Prediction** is a problem defined on TGs aiming to predict a property (usually existence) of a link between a pair of nodes at a future timestamp. In our work we focus on this problem, as we believe it is of fundamental nature for quantifying the behavior of models.

**Negative sampling** is a method commonly used to train and evaluate TG methods. Poursafaei et al. (2022) identify weaknesses in widely used uniform random sampling and propose to sample *historical negatives* - past edges absent in the current time step for TGs. TGB also employ this strategy in their evaluation protocol.

**Datasets** we use for our experiments include: `tgbl-wiki` (small) a network of editors editing Wikipedia pages, `tgbl-review` (small) a network of users rating Amazon products, `tgbl-coin` (medium) a network of cryptocurrency transactions, `tgbl-comment` (large) a network of Reddit users replying to other users. For further details on datasets we refer to Huang et al. (2023). Since

Table 1: Percentage of perfect 1.0 scores for top K destination nodes with most interactions in previous N interactions. Values improved by RP-NS are underscored. * denotes our contribution.

| K | N | TGN | | DyRep | | TGN+RP-NS* | | DyRep+RP-NS* | |
|---|---|---------|------|---------|------|---------|------|---------|------|
| | | comment | coin | comment | coin | comment | coin | comment | coin |
| 50 | 5000 | 90.10% | 61.87% | 43.35% | 5.83% | 2.08% | 8.47% | 0% | 7.83% |
| 100 | 5000 | 86.46% | 57.57% | 43.99% | 5.39% | 2.18% | 7.52% | 0% | 5.58% |
| 1000 | 5000 | 69.55% | 23.15% | 45.34% | 4.00% | 3.05% | 2.82% | 0% | 1.11% |
| 50 | 20000 | 92.0% | 62.40% | 43.90% | 5.90% | 2.19% | 8.40% | 0% | 7.94% |
| 100 | 20000 | 88.40% | 58.42% | 44.60% | 5.23% | 2.27% | 7.54% | 0% | 5.85% |
| 1000 | 20000 | 71.67% | 32.86% | 45.87% | 5.15% | 3.14% | 3.85% | 0% | 1.42% |
| 50 | 100000 | 87.00% | 62.79% | 42.37% | 5.98% | 2.10% | 8.56% | 0% | 8.07% |
| 100 | 100000 | 84.75% | 59.88% | 44.05% | 5.24% | 2.38% | 7.73% | 0% | 5.94% |
| 1000 | 100000 | 71.10% | 39.82% | 46.89% | 5.68% | 3.55% | 4.47% | 0% | 1.66% |

publication of TGB benchmark, `tgbl-wiki` and `tgbl-review` have been modified. We report results on both versions: *v1* originally reported in Huang et al. (2023) and *v2* from TGB Website (2023). All dynamic link property prediction problems on these datasets involve predicting the existence of an edge.

**Our contributions:** We build upon the latest available version of TGB as of the time of writing (`0.8.0`), to further analyze and improve training and evaluation methods for TGs. We identify a strikingly simple and effective baseline that shows inadequacies of current training and evaluation protocols. We propose improved negative sampling protocols for training and evaluation and demonstrate their effectiveness. We identify weaknesses in existing TG models on a class of datasets with strong global dynamics. We introduce efficient measures of global dynamics strength for TGs allowing a better understanding of *how temporal* a TG dataset is. We conduct a comparison with a non-contrastive method and report its superiority.

**Replicability:** Our anonymized code is available at: github.com/temporal-graphs-negative-sampling/TGB

## 2 OBSERVATIONS OF PERFECTLY SATURATED SCORES

By analyzing predictions of TGN (Rossi et al., 2020) and DyRep (Trivedi et al., 2019) models we find that *recently globally popular destination nodes* have frequently oversaturated scores (perfectly equal to 1.0). We define the class formally, as top $K$ destination nodes with the most interactions, in the previous $N$ interactions in the temporal graph. We report exact percentages of oversaturated scores for different $K$ and $N$ in Table 1.

Perfect 1.0 scores cannot be distinguished and their relative ranking is uninformative. Additionally, the identified class of oversaturated nodes may inspire a good baseline model. Before we address these observations, we will measure the degree of informativeness of recent popularity in datasets.

## 3 MEASURES OF GLOBAL TEMPORAL DYNAMICS IN DATASETS

We wish to measure how much information recent global node popularity provides for future edges in a temporal graph dataset, a type of autocorrelation on temporal graphs. As a reasonable simplification, we model a dataset's destination nodes as the result of a discrete stochastic process, where at every timestep $T_i$, $K$ samples are independently drawn from some categorical distribution $P[T_i]$. For efficiency purposes, and to ensure an equal comparison we set $K$ for each dataset, so that it is divided into exactly $N$ timesteps $T_i$. The normalized counts of destination nodes at time step $T_i$ yield $Q[T_i]$, which serves as an empirical approximation of the underlying categorical distribution's probability mass function (PMF). To compare these PMFs at different time steps $T_i$ we employ the $W_1$ Wasserstein Metric, also known as the Earth Mover's Distance.

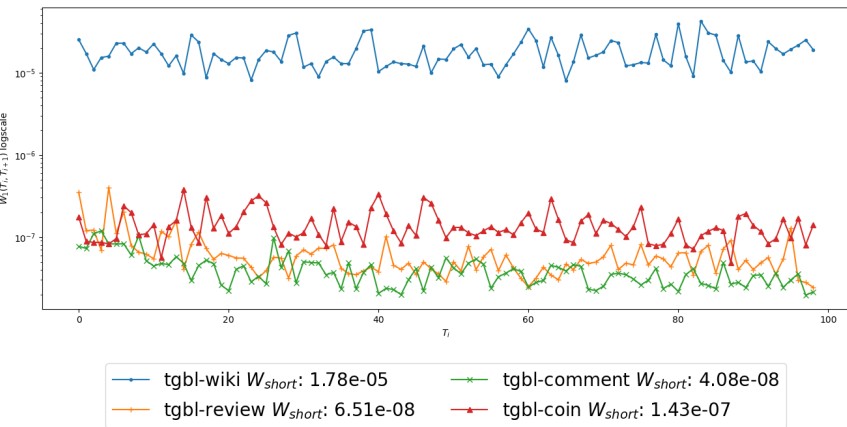

Figure 1: Short-horizon measure of global temporal dynamics $W_{short}$ with $N = 100$ timesteps.

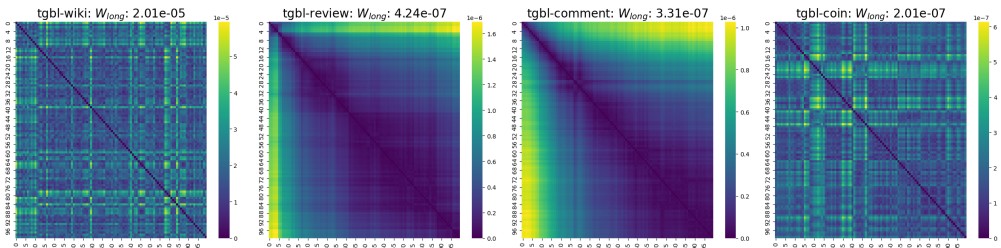

Figure 2: Long-range measure of global temporal dynamics $W_{long}$ with $N = 100$ timesteps.

### 3.1 A SHORT-HORIZON MEASURE OF GLOBAL DYNAMICS IN DATASETS

To measure how much the most recent historical information can inform future timesteps in a dataset's evolution, we can calculate the distances of neighboring timesteps sequentially. We propose the following measure, where $W_1$ denotes the Earth Mover's Distance:

$$W_{short} = \frac{1}{N} \cdot \sum_{i=0}^{N-1} W_1(Q[T_i], Q[T_{i+1}])$$

The lower this value, the more informative are historical global node popularities for the immediately next timestep. We report the measure's results, as well as plot all the individual distances for all datasets in Figure 3.1. It can be observed that `tgbl-wiki` has the highest measure, implying that global temporal correlations are low. This can likely be attributed to a lack of major social effects on Wikipedia, compared to the remaining three datasets: `tgbl-comment`, `tgbl-coin` and `tgbl-review`, where user behaviors are more likely to be driven by trends, hype cycles or seasonality.

### 3.2 A LONG-RANGE MEASURE OF GLOBAL DYNAMICS IN DATASETS

The $W_{short}$ measure captures only short-term effects. We may also be interested in a longer time-range of influence. We can extend $W_{short}$ to a mid and long-term context with:

$$W_{long} = \frac{N \times (N-1)}{2} \cdot \sum_{i=0}^{N} \sum_{j=0}^{i-1} W_1(Q[T_i], Q[T_j])$$

A low $W_{long}$ value indicates strong medium-term or even long-term stability of the node frequency distribution. We report results, as well as plot heat maps of long-term distances in Figure 3.1. From an inspection of the heat maps, it becomes apparent, that `tgbl-comment` and `tgbl-review` data generating processes follow a very smooth long-horizon evolution. The behavior of `tgbl-wiki` is chaotic with no obvious patterns, `tgbl-coin` forms a checkered pattern – possibly some abrupt switching between different dominant modes of behavior with sudden reversals.

# 4 UNEXPECTEDLY STRONG BASELINE FOR DYNAMIC LINK PROPERTY PREDICTION

Based on the observation that *recently globally popular destination nodes* can be good candidates, we construct an extremely simple and efficient baseline, we call ***PopTrack*** (Popularity Tracking). The algorithm works sequentially in batches on a time-ordered temporal graph dataset. It maintains a time-decayed vector $P$ of occurrence counters for all destination nodes. Its hyperparameters are: $batchsize$ and decay factor $\lambda$. For prediction, the state of $P$ from the previous batch is taken as node scores. Afterwards, the counter is incremented with all destination nodes from the current batch and multiplicative decay $\lambda$ is applied to all entries in $P$. The baseline is a type of exponential smoothing of destination node popularity, its pseudocode is shown in Algorithm 1.

---

**Algorithm 1:** PopTrack: A temporal popularity baseline for dynamic link property prediction

---

**Data:** $dataloader$ - temporally ordered sequence of batches, $\lambda$ - decay factor,

**(1)** $P := Vect[num\_nodes](0)$;
**(2)** **foreach** $source\_nodes, destination\_nodes \in dataloader$ **do**
**(3)** $\quad$ $predictions \leftarrow TopK(P)$;
**(4)** $\quad$ **foreach** $dst \in destination\_nodes$ **do**
**(5)** $\quad\quad$ $P[dst] \leftarrow P[dst] + 1$;
**(6)** $\quad$ $P \leftarrow P \cdot \lambda$;

---

## 4.1 PERFORMANCE OF THE POPTRACK BASELINE

We test Algorithm 1 on all dynamic link property prediction datasets, except for `tgbl-flight`, which is broken in TGB version `0.8.0` at the time of writing. We hold the $batchsize$ fixed at 200 and perform a grid search for optimal $\lambda$ for each dataset on its validation set. Our method is fully deterministic.

We establish new state-of-the-art results on `tgbl-comment` ($\lambda = 0.96$), `tgbl-coin` ($\lambda = 0.94$) and `tgbl-review-v1` ($\lambda = 0.999$), take the 2nd place for `tgbl-review-v2` ($\lambda = 0.999$). Our baseline outperforms a surprisingly large and diverse number of neural temporal graph models, as well as the EdgeBank heuristics. We only note negative results on `tgbl-wiki` ($\lambda = 0.38$) datasets, which are outliers with very high $W_{short}$ and $W_{long}$ measures, indicating weak global temporal dynamic effects. We report detailed results in Table 2. Our results imply that existing TG models fail to learn global temporal dynamics.

The baseline's runtime is below 10 minutes on all datasets on a single CPU, with negligible memory usage (only needing to store a single number for each node), in stark contrast to neural approaches requiring days of training on powerful GPUs (Huang et al., 2023).

## 4.2 ANALYSIS AND INTUITION

We have shown that a simply tracking *recent global popularity* of destination nodes is a very strong baseline on many dynamic link prediction tasks. The strength of the baseline is somewhat surprising, especially given the immense expressive power of neural approaches it outperforms. This can be caused either by inadequacy of existing graph models to capture rapidly changing, global temporal dynamics, the way they are trained, or a mix of both.

Table 2: Results for dynamic link property prediction task on various datasets from Huang et al. (2023) and TGB Website (2023). Best results are underlined and bold, second best are bold. * denotes our contribution.

| Method | tgbl-coin | | tgbl-comment | |
| --- | --- | --- | --- | --- |
| | Validation MRR | Test MRR | Validation MRR | Test MRR |
| DyRep (Trivedi et al., 2019) | $0.507 \pm 0.029$ | $0.434 \pm 0.038$ | $0.291 \pm 0.028$ | $0.289 \pm 0.033$ |
| TGN (Rossi et al., 2020) | $\mathbf{0.594 \pm 0.023}$ | $\mathbf{0.583 \pm 0.050}$ | $\mathbf{0.356 \pm 0.019}$ | $\mathbf{0.379 \pm 0.021}$ |
| EdgeBank$_{tw}$ (Poursafaei et al., 2022) | 0.492 | 0.580 | 0.1244 | 0.1494 |
| EdgeBank$_{\infty}$ (Poursafaei et al., 2022) | 0.315 | 0.359 | 0.1087 | 0.1285 |
| PopTrack* | **_0.715_** | **_0.725_** | **_0.6903_** | **_0.729_** |

| Method | tgbl-review-v1 | | tgbl-review-v2 | |
| --- | --- | --- | --- | --- |
| | Validation MRR | Test MRR | Validation MRR | Test MRR |
| GraphMixer (Cong et al., 2023) | $0.411 \pm 0.025$ | $0.514 \pm 0.020$ | $\mathbf{0.428 \pm 0.019}$ | $\mathbf{0.521 \pm 0.015}$ |
| TGAT (Xu et al., 2020) | - | - | $0.324 \pm 0.006$ | $0.355 \pm 0.012$ |
| TGN (Rossi et al., 2020) | $\mathbf{0.465 \pm 0.010}$ | $\mathbf{0.532 \pm 0.020}$ | $0.313 \pm 0.012$ | $0.349 \pm 0.020$ |
| NAT (Luo & Li, 2022) | - | - | $0.302 \pm 0.011$ | $0.341 \pm 0.020$ |
| DyGFormer (Yu, 2023) | - | - | $0.219 \pm 0.017$ | $0.224 \pm 0.015$ |
| DyRep (Trivedi et al., 2019) | $0.356 \pm 0.016$ | $0.367 \pm 0.013$ | $0.216 \pm 0.031$ | $0.220 \pm 0.030$ |
| CAWN (Wang et al., 2022) | $0.201 \pm 0.002$ | $0.194 \pm 0.004$ | $0.200 \pm 0.001$ | $0.193 \pm 0.001$ |
| TCL (Wang et al., 2021) | $0.194 \pm 0.012$ | $0.200 \pm 0.010$ | $0.199 \pm 0.007$ | $0.193 \pm 0.009$ |
| EdgeBank$_{tw}$ (Poursafaei et al., 2022) | 0.0894 | 0.0836 | 0.024 | 0.025 |
| EdgeBank$_{\infty}$ (Poursafaei et al., 2022) | 0.0786 | 0.0795 | 0.023 | 0.023 |
| PopTrack* | **_0.470_** | **_0.549_** | **0.341** | **0.414** |

| Method | tgbl-wiki-v1 | | tgbl-wiki-v2 | |
| --- | --- | --- | --- | --- |
| | Validation MRR | Test MRR | Validation MRR | Test MRR |
| GraphMixer (Cong et al., 2023) | $0.707 \pm 0.014$ | $0.701 \pm 0.014$ | $0.113 \pm 0.003$ | $0.118 \pm 0.002$ |
| TGAT (Xu et al., 2020) | - | - | $0.131 \pm 0.008$ | $0.141 \pm 0.007$ |
| TGN (Rossi et al., 2020) | $\mathbf{0.737 \pm 0.004}$ | $\mathbf{0.721 \pm 0.004}$ | $0.435 \pm 0.069$ | $0.396 \pm 0.060$ |
| NAT (Luo & Li, 2022) | - | - | $\mathbf{0.773 \pm 0.011}$ | $\mathbf{0.749 \pm 0.010}$ |
| DyGFormer (Yu, 2023) | - | - | **_0.816 \pm 0.005_** | **_0.798 \pm 0.004_** |
| DyRep (Trivedi et al., 2019) | $0.411 \pm 0.015$ | $0.366 \pm 0.014$ | $0.072 \pm 0.009$ | $0.050 \pm 0.017$ |
| CAWN (Wang et al., 2022) | $\mathbf{0.794 \pm 0.014}$ | $\mathbf{0.791 \pm 0.015}$ | $0.743 \pm 0.004$ | $0.711 \pm 0.006$ |
| TCL (Wang et al., 2021) | $0.734 \pm 0.007$ | $0.712 \pm 0.007$ | $0.198 \pm 0.016$ | $0.207 \pm 0.025$ |
| EdgeBank$_{tw}$ (Poursafaei et al., 2022) | 0.641 | 0.641 | 0.600 | 0.571 |
| EdgeBank$_{\infty}$ (Poursafaei et al., 2022) | 0.551 | 0.538 | 0.527 | 0.495 |
| PopTrack* | 0.538 | 0.512 | 0.105 | 0.097 |

The performance of our baseline compared to other methods strongly correlates with the measures of global dynamics $W_{short}$ and $W_{long}$. The sources of global dynamics are easy to pinpoint by analyzing the data generating processes of the datasets themselves. For instance, `tgbl-comment` dataset consists of edges generated by (source) Reddit users responding to (destination) Reddit users' posts. The nature of Reddit and other social networks is such that highly engaging content is pushed to the top of the website (or particular *subreddits*), where it further benefits from high observability, in a self-reinforcing cycle. The active lifetime of a piece of content is usually measured in hours or days at most. After this period, the content loses visibility, becomes harder to discover and harder to engage with.

The phenomenon of short-lived, self-reinforcing popularity is present in other areas of digital social life such as X (Twitter), Facebook, and even e-commerce stores (*fast fashion trends*), cryptocurrency trading activities (*hype cycles*). It is worth noting that users may have different tastes and interests and be exposed to different subsets of currently popular information with varying dynamics. E.g. a person interested in `/r/Mathematics` subreddit, may be exposed to lower-paced content, than someone tracking `/r/worldnews`. A global baseline is unable to track those local effects, but it's an interesting avenue for future research.

# 5 TOWARDS A MORE RELIABLE EVALUATION METHOD FOR DYNAMIC LINK PREDICTION

We revisit the problem hinted at by results in Table 1. If up to $90\%$ scores for the recent top 50 destination nodes are perfect $1.0$ in a TGN model, they cannot be ordered meaningfully. Our simple baseline PopTrack, despite achieving a very high MRR, disregards the source node context, returning the same predictions for all nodes at a given timestep. This might imply that the evaluation protocol in Huang et al. (2023) is insufficient to reflect real-world usefulness of models. The ability to both *accurately rank recently popular destination nodes*, as well as to *vary predictions depending on the source node*, seem to be reasonable requirements for a good temporal graph model.

## 5.1 THE CURRENT EVALUATION METHOD

The evaluation protocol proposed in Huang et al. (2023) consists of sampling 20 negative edges for validation and testing. The authors introduce two methods of choosing negative edges: *historical* and *random*, where *historical* are edges previously observed in the training set, but not at the current timestep, and *random* are just random. Both methods are utilized equally. We call this original metric $MRR_{naive}$.

In datasets with strong non-stationary dynamics, there is a high probability that most of the negative examples are *stale* (they do not belong to the class of recently popular destination nodes), while only the positive sample is *fresh*, thus *hard* negative candidates are rarely observed.

## 5.2 AN IMPROVED EVALUATION METHOD

To bring evaluation results closer to real-world usefulness, we propose an improved evaluation method, by sampling from *top N recently popular items according to PopTrack* model in addition to the original method proposed by Huang et al. (2023). Sampling e.g. 20 items from top 1000 recently most-popular destination nodes, 5 items from *historical* edges and 5 *random* edges would constitute a reasonable blend of methods. The combined approach remediates the lack of *hard* candidates, but still ensures that *easy* candidates are scored correctly. Since results for the original $MRR_{naive}$ are already known, in our research we focus on benchmarking with pure *top N* part. With thoroughness in mind, we perform a full MRR evaluation on all top 20, top 100 and top 500 recently most popular candidates without sampling, denoted as $MRR_{top20}$, $MRR_{top100}$ and $MRR_{top500}$ respectively. Similarly to Huang et al. (2023) we generate fixed lists of negative samples, to ensure reproducibility and consistency when comparing across models.

We perform the evaluations on the 2 largest datasets with all heuristic models (EdgeBank variants and ours) and the two most popular graph neural models: DyRep and TGN. Full evaluation quickly becomes expensive, so we limit our analysis to two models and maximal $N = 500$. We report results in Table 3. On all $MRR_{topN}$ metrics, our PopTrack baseline performs poorly – as intended – proving that the metrics behave differently than $MRR_{naive}$. Both EdgeBank methods perform decently well, but performance of TGN and DyRep models is very lacking. While they were somewhat good in discriminating *hard* candidates from *easy* ones, they fail to rank *hard* candidates properly. Compared to EdgeBank baselines, their performance drops more significantly as $N$, the number of top candidates, grows. This implies that $MRR_{naive}$ is a poor approximation of full MRR.

# 6 AN IMPROVED NEGATIVE SAMPLING SCHEME FOR TRAINING

Having improved the evaluation metric, we will now propose improvements to the training protocol. Score oversaturation problems observed in Table 1 likely arise due to the sparsity of *hard* negative candidates during training. The model training protocol employed in Huang et al. (2023) involves uniformly randomly sampling a single negative edge with no temporal awareness.

## 6.1 NEGATIVE SAMPLES IN NON-STATIONARY ENVIRONMENTS

For temporal graphs, the topic of negative sampling is largely unexplored, with the most recent findings by Poursafaei et al. (2022). The authors introduce the *historical* way of negative sampling,

which is already utilized by the TGB Benchmark in Huang et al. (2023) and as we have demonstrated is insufficient to achieve good results.

Temporal graphs with non-stationary node popularity distributions pose an additional challenge, which is not captured by prior methods. Node popularity distribution evolves over time and it becomes necessary to track these shifts to generate high quality *hard* negative samples. To remedy this issue, we propose an improved negative sampling scheme for dynamic link property prediction on temporal graphs called *Recently Popular Negative Sampling* (RP-NS).

## 6.2 METHOD

We introduce *Recently Popular Negative Sampling* (RP-NS) based on PopTrack. Instead of sampling negative destination nodes uniformly, we sample 90% of candidates from a popularity distribution given by our simple baseline, to the power of $\frac{3}{4}$ (both numbers chosen empirically). The remaining 10% of candidates are sampled uniformly, to ensure that the model sees both *hard* and *easy* candidates during training.

## 6.3 RESULTS

Results of TGN and DyRep training on `tgbl-coin` and `tgbl-comment` with our RP-NS scheme are reported in Table 3. We report both the original $MRR_{naive}$ metric as well as our additional *hard candidate* metrics. We observe not only comparable or better results for $MRR_{naive}$, but also significantly improved results for $MRR_{topN}$ for both models on both datasets. Degradation of $MRR_{topN}$ as $N$ grows is still substantial for both neural models and may require changes to their architectures to be remediated fully.

Table 3: Comparison of models trained naively and with Recently Popular Negative Sampling (RP-NS) under naive and topN evaluation schemes. * denotes our contribution.

| Method | tgbl-coin | | tgbl-comment | |
|---|---|---|---|---|
| | Val $MRR_{naive}$ | Test $MRR_{naive}$ | Val $MRR_{naive}$ | Test $MRR_{naive}$ |
| DyRep (Trivedi et al., 2019) | 0.507 | 0.434 | 0.291 | 0.289 |
| DyRep (Trivedi et al., 2019) + RP-NS* | 0.469 | 0.469 | 0.390 | 0.404 |
| TGN (Rossi et al., 2020) | 0.594 | 0.583 | 0.356 | 0.379 |
| TGN (Rossi et al., 2020) + RP-NS* | 0.592 | 0.546 | 0.441 | 0.393 |
| EdgeBank$_{tw}$ (Poursafaei et al., 2022) | 0.492 | 0.580 | 0.1244 | 0.1494 |
| EdgeBank$_{\infty}$ (Poursafaei et al., 2022) | 0.315 | 0.359 | 0.1087 | 0.1285 |
| PopTrack* | **0.715** | **0.725** | **0.6903** | **0.729** |
| EMDE (Dąbrowski et al., 2021) (non-contrastive)* | 0.703 | 0.674 | 0.455 | 0.426 |
| | Val $MRR_{top20}$* | Test $MRR_{top20}$* | Val $MRR_{top20}$* | Test $MRR_{top20}$* |
| DyRep (Trivedi et al., 2019) | 0.224 | 0.226 | 0.128 | 0.126 |
| DyRep (Trivedi et al., 2019) + RP-NS* | 0.175 | 0.150 | 0.233 | 0.236 |
| TGN (Rossi et al., 2020) | 0.103 | 0.103 | 0.086 | 0.088 |
| TGN (Rossi et al., 2020) + RP-NS* | **0.510** | 0.453 | **0.336** | **0.329** |
| EdgeBank$_{tw}$ (Poursafaei et al., 2022) | 0.487 | 0.535 | 0.213 | 0.211 |
| EdgeBank$_{\infty}$ (Poursafaei et al., 2022) | 0.509 | **0.554** | 0.212 | 0.211 |
| PopTrack* | 0.117 | 0.113 | 0.066 | 0.065 |
| EMDE (Dąbrowski et al., 2021) (non-contrastive)* | **0.630** | **0.601** | **0.364** | **0.3391** |
| | Val $MRR_{top100}$* | Test $MRR_{top100}$* | Val $MRR_{top100}$* | Test $MRR_{top100}$* |
| DyRep (Trivedi et al., 2019) | 0.088 | 0.081 | 0.029 | 0.029 |
| DyRep (Trivedi et al., 2019) + RP-NS* | 0.089 | 0.064 | 0.097 | 0.096 |
| TGN (Rossi et al., 2020) | 0.027 | 0.027 | 0.019 | 0.019 |
| TGN (Rossi et al., 2020) + RP-NS* | 0.341 | 0.277 | **0.150** | **0.118** |
| EdgeBank$_{tw}$ (Poursafaei et al., 2022) | 0.374 | 0.414 | 0.110 | 0.113 |
| EdgeBank$_{\infty}$ (Poursafaei et al., 2022) | **0.391** | **0.423** | 0.106 | 0.109 |
| PopTrack* | 0.092 | 0.088 | 0.032 | 0.031 |
| EMDE (Dąbrowski et al., 2021) (non-contrastive)* | **0.557** | **0.525** | **0.248** | **0.225** |
| | Val $MRR_{top500}$* | Test $MRR_{top500}$* | Val $MRR_{top500}$* | Test $MRR_{top500}$* |
| DyRep (Trivedi et al., 2019) | 0.018 | 0.018 | 0.006 | 0.006 |
| DyRep (Trivedi et al., 2019) + RP-NS* | 0.045 | 0.029 | 0.042 | 0.040 |
| TGN (Rossi et al., 2020) | 0.010 | 0.009 | 0.004 | 0.004 |
| TGN (Rossi et al., 2020) + RP-NS* | 0.147 | 0.081 | **0.058** | 0.030 |
| EdgeBank$_{tw}$ (Poursafaei et al., 2022) | 0.302 | 0.324 | 0.057 | **0.061** |
| EdgeBank$_{\infty}$ (Poursafaei et al., 2022) | **0.314** | **0.334** | 0.054 | 0.057 |
| PopTrack* | 0.088 | 0.083 | 0.026 | 0.025 |
| EMDE (Dąbrowski et al., 2021) (non-contrastive)* | **0.491** | **0.468** | **0.199** | **0.180** |
| | Val $MRR_{all}$* | Test $MRR_{all}$* | Val $MRR_{all}$* | Test $MRR_{all}$* |
| EMDE (Dąbrowski et al., 2021) (non-contrastive)* | **0.407** | **0.390** | **0.134** | **0.1210** |

We also compare the level of scores oversaturation, which have initially motivated us to investigate the problems. Results for models trained with the improved scheme are reported in Table 1. Comparing the results to the values without RP-NS, we can see that the number of oversaturated scores drops significantly across models and datasets. The improvements are very notable, but a small level of oversaturation persists - an opportunity for future work on improving model architectures.

## 7 ALTERNATIVES TO NEGATIVE SAMPLING

The number of nodes in the benchmark datasets is very large (up to 1 million), so both training and evaluation with negative sampling seem justified. Nonetheless, we can see from the rapid degradation of $MRR_{topN}$ as $N$ grows, that such evaluations may be a poor proxy for a full $MRR$ calculation. Methods which can be trained non-contrastively at scale exist. Efficient Manifold Density Estimator (EMDE) (Dąbrowski et al., 2021) is one such model, combining the idea of Count-Sketches with locality-sensitive hashes computed on static node embeddings. It approximates an extremely wide Softmax output with multiple independent ones, like a Bloom Filter approximates one-hot encoding.

### 7.1 EXPERIMENT RESULTS

We train EMDE, generating embeddings with Cleora (Rychalska et al., 2021), an unsupervised node embedding method, with 6 iterations on the training set, resulting in 70 locality-sensitive hashes and 20 random hashes of cardinality 512. This results in initial node representations of depth 90 and width 512. To create temporal input to a neural network, for every source node we aggregate its incoming edges from historic batches with a rolling decay factor of 0.7, and apply the same procedure to outgoing edges, obtaining 2 sketches which are flattened and concatenated. Targets are initial representations of the destination nodes (interpreted as single-element Bloom Filters) without any temporal aggregation. The neural network has a hidden size of 4000 and consists of 6 layers with LeakyReLU activations and Layer Normalization applied post-norm. Total number of parameters of the network is $\sim 750$M, majority taken up by in/out projection matrices. We train with AdamW optimizer for 3 epochs with $1e-4$ learning rate and a batch size of 512.

We report results in Table 3. We note that not only does EMDE outperform the other methods on both datasets, but its lead grows as $N$ grows for $MRR_{topN}$ evaluation. Thanks to a static representation of target candidates, it is the only method for which we are able to efficiently perform (within hours) a full $MRR_{all}$ evaluation on all possible destination nodes. For TGN or DyRep, a full evaluation in the TGB setup would take more than a calendar year. We observe that for EMDE the results of $MRR_{top500}$ and $MRR_{full}$ do not differ much, despite maximum values of $N$ being well over $500,000$ for both datasets. While unknown, it seems unlikely that the same would hold for TGN and DyRep, given how sharply their scores decline as $N$ grows.

## 8 LIMITATIONS

We were unable to perform experiments on `tgbl-flight` which is broken at the time of writing in TGB version `0.8.0`, pending an unresolved Github issue. Due to extreme compute requirements of graph methods, we limited our experiments with models trained from scratch to the most popular ones: TGN and DyRep and two most challenging datasets: `tgbl-comment` and `tgbl-coin`. Calculation of $MRR_{full}$ for any of the neural graph models is prohibitively expensive, because candidate representations need to be dynamically built for all evaluation timestamps and all candidates. Nonetheless, we believe that our observations are valid for a broader class of datasets and neural architectures. We invite the research community to extend our experiments to other temporal graph network architectures.

## 9 CONCLUSION

Our results prove an insufficiency of prior metrics for TG models, being easily beaten by PopTrack – our simple baseline on datasets with strong global temporal effects. Measures $W_{short}$ and $W_{long}$ allow a quick estimation of temporal graph datasets' autocorrelation. Improved MRR metrics we propose are more robust and better capture the true performance of models. We show that negative

sampling during training of TG models is a subtle problem and propose improvements which deliver improved results for the new metrics. Most importantly, we show that existing TG models trained with negative sampling have problems with capturing global temporal dynamics on strongly dynamic datasets, and their evaluation on negative examples may be an insufficient proxy for a full evaluation. A comparison with a fundamentally different approach trained non-contrastively shows it to be more suitable for real-world scenarios. We hope that our contributions will allow more robust and reliable evaluation of TG models and inspire new model architectures.

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
