# OpenReview forum: "Temporal graph models fail to capture global temporal dynamics"
_ICLR.cc/2024/Conference — ICLR 2024 Conference Withdrawn Submission_

### Official Review · Reviewer_C9Fh · 2023-10-29

**Soundness:** 1 poor
**Presentation:** 1 poor
**Contribution:** 1 poor
**Rating:** 3
**Confidence:** 4

**Summary:**

This paper makes an observation that a trivial optimization-free baseline of "recently popular nodes" outperforms other methods on medium and large-size datasets in the Temporal Graph Benchmark (TGB). Two measures based on Wasserstein distance which can quantify the strength of short-term and long-term global dynamics of datasets are proposed. By evaluating the performance of Temporal Graph baselines on several datasets, the authors try to reveal that temporal graph network architectures need deep rethinking for usage in problems with significant global dynamics.

**Strengths:**

This paper makes an interesting observation that sometimes a trivial model can outperform the state-of-the-art Temporal Graph models on certain datasets.

**Weaknesses:**

1. This paper is poorly written and difficult to read. The organization of the sections is very confusing.  The reviewer can hardly follow the idea of this paper.

2. The figures in this paper do not seem to provide any useful information. It is hard to tell what figure 1 and figure 2 try to show to the reader and they are not referenced or explained anywhere in the paper.

3. The main claim in this paper, which is the proposed PopTrack method "beat" other TG models, is hardly supported by empirical evaluations. In table 2, PopTrack performs significantly worse than the best baseline on 3 out of 6 datasets. On the first two datasets where PopTrack dominates, it is only compared with 4 baselines instead of 10 baselines like others, and the reason behind this is not explained in the paper.

4. The literature review is far from enough. Only 13 papers are referenced. Necessary background information and preliminaries are missing (e.g., the definitions of dynamic graphs; the introduction of existing TG models, etc.)

5. All experiments are conducted on the specific TGB datasets, which may follow certain patterns that can be easily captured by a simple model as “shortcuts”. It is unclear to what extent the conclusions drawn from here still hold on other sources of datasets, which narrows its applicability and limits the contributions of the paper.

**Questions:**

See weaknesses.

---

### Official Review · Reviewer_mQmi · 2023-10-30

**Soundness:** 1 poor
**Presentation:** 1 poor
**Contribution:** 2 fair
**Rating:** 1
**Confidence:** 2

**Summary:**

In this paper, the authors focused on the dynamic link property prediction task on the well-known Temporal Graph Benchmark (TGB) and tried to demonstrate the limitations of existing baselines and evaluation protocols in TGB via a great amount of experiments. New measures and negative sampling schemes were proposed to tackle these limitations. The authors also provided their code for review.

**Strengths:**

S1. The authors have conducted a great amount of experiments regarding dynamic link property prediction on the well-known Temporal Graph Benchmark (TGB).

S2. The author provided the code regarding their experiments for review.

**Weaknesses:**

**W1. There are no formal statements regarding the problem/task considered in this paper. Some statements are also inconsistent.**

For instance, it is unclear that the authors considered (1) which data model (e.g., discrete-time dynamic graph or continuous-time dynamic graph) of dynamic graphs, (2) whether graph attributes (e.g., node and edge attributes) are available, (3) whether the edges are directed or undirected, (4) whether the edges are unweighted or weighted, etc.

What is the formal definition of dynamic link property prediction? Does it only refer to dynamic/temporal link prediction (formulated as the binary edge/node-pair classification) as negative sampling is usually adopted for link prediction? In contrast, dynamic link property prediction can sometimes be formulated as multi-class edge/node-pair classification.

In Table 1, how to compute the 'top $K$ destination nodes with the most interactions in the previous $N$ interactions in the temporal graph'? It is recommended to add a simple toy example or pseudo-code to demonstrate such a procedure.

As I known, temporal/dynamic link prediction usually includes the transductive and inductive settings. It is also unclear which setting(s) did the authors consider in this study.

***

**W2. The intuitions and motivations of some significant statements are weak or unclear.**

In Section 2, why the percentage of perfect 1.0 scores for top $K$ destination nodes with most interactions in previous $N$ interactions can indicates that 'recently globally popular destination nodes have frequently oversaturated scores'? What does the 'score' refer to? It is suggested to add some more intuitive interpretations.

Global temporal dynamics is a significant concept of this study. However, after reading the whole paper, I still fail to understand what is global temporal dynamics due to the lack of formal definitions and intuitive examples. What kinds of features can be defined as global temporal dynamics? Are there any concrete examples in real dynamic graphs? Accordingly, what kinds of features can be defined as local temporal dynamics?

Why Wasserstein metric (i.e., Earth Mover’s distance) can be used to measure global temporal dynamics (e.g., based on what properties of this metric)? The formal definition of this metric is also not given.

In the 1st paragraph of Section 4, why we can reach the conclusion that 'recently globally popular destination nodes can be good candidates' (e.g., compared with what)?

In summary, due to the aforementioned issues, I still cannot understand why 'temporal graph models fail to capture global temporal dynamics' as stated in the tile of this paper.

***

**W3. The overall presentation is poor.**

In Table 1, it is unclear what do the underlined results indicate.

In the 2nd paragraph of Page 3 and the 1st paragraph of Page 4, it is unclear what does 'Figure 3.1' refer to. There are no Figure 3.1 in the paper.

The definitions of the numbers in Fig. 1 (e.g., 1.78e-05, 4.08e-08, etc.) are not given. It is also similar for Fig. 2. The text of the x- and y-axis in Fig. 1 and 2 are too small, which are hard to read.

In Table 2, why there are some missing results (denoted as '-')?

In the caption of Table 2, the authors claimed that the 'best results are underlined and bold, second best are bold'. It seems that there is a minor error. It is unclear whether the best results are underlined or in bold.

The term 'dynamic link property prediction' is used in the title of Section 4 but 'dynamic link prediction' is used in the title of Section 5, which seems to be inconsistent.

In Table 3, it is unclear what do the underlined and bold results indicate.

There are no literature reviews (e.g., usually in an independent section) regarding existing related work.

**Questions:**

See W1-W3.

As I can check, the authors only considered the temporal/dynamic link prediction on unweighted graphs. In addition, there are some other studies considered the advanced prediction for weighted future links [1-4], which should not only determine the existence of a future link but also the corresponding link weight. Can the proposed method be extended to handle such an advanced settings?

[1] GCN-GAN: A Non-linear Temporal Link Prediction Model for Weighted Dynamic Networks. IEEE InfoCOM, 2019.

[2] An Advanced Deep Generative Framework for Temporal Link Prediction in Dynamic Networks. IEEE IEEE Transactions on Cybernetics, 2020.

[3] High-Quality Temporal Link Prediction for Weighted Dynamic Graphs via Inductive Embedding Aggregation. IEEE TKDE, 2023.

[4] Temporal link prediction: A unified framework, taxonomy, and review. ACM Computing Surveys, 2023.

---

### Official Review · Reviewer_Chai · 2023-10-31

**Soundness:** 3 good
**Presentation:** 3 good
**Contribution:** 3 good
**Rating:** 6
**Confidence:** 3

**Summary:**

This paper shows that for the dynamic link property prediction
of temporal graphs, a simple optimization-free baseline PopTrack, which
just tracks recently popular nodes, outperforms other methods in the
literature on medium and large-size datasets in the Temporal Graph
benchmark.  It then proposes a new negative sampling scheme as well as
a new evaluation method based on the PopTrack.  It also proposes a
method based on the efficient manifold density estimator that utilizes
locality-sensitive hashing and does not rely on negative sampling.

**Strengths:**

- By using the simple baseline PopTrack, the authors reveal the insufficiency of the conventional methods and metrics for TG models.
- The authors propose an alternative metric that better captures the true performance of models.
- They also propose a new negative sampling method and a new non-contrastive method.
- This paper is clearly written with plain language. Their findings and the proposed new metric enhance the value of the Temporal Graph benchmark and may help accelerate the TG research.

**Weaknesses:**

- The proposed non-contrastive EMDE method may be time consuming.
- The effectiveness of the proposed negative sampling method is still limited.

**Questions:**

The non-contrastive EMDE method shows a good performance but its scalability is unclear. I would like to know how the method is scalable. While the runtime of the PopTrack baseline is reported as below 10 minutes, the runtime of the other methods are not reported. I would like to know them.

In Algorithm 1, the step (3) is confusing. It looks as if $predictions$ is used to update $P$ but it is not.
I think the step (3) should be placed outside of the foreach loop.

They should clarify the definition fo MRR. I assume MRR is mean reciprocal rank.

**Details Of Ethics Concerns:**

I have no concerns.

---

### Official Review · Reviewer_BqDA · 2023-11-03

**Soundness:** 2 fair
**Presentation:** 2 fair
**Contribution:** 2 fair
**Rating:** 3
**Confidence:** 4

**Summary:**

This paper is a study on a recent published dataset benchmark on temporal graphs. The authors analyze the benchmark dataset with some comparison methods on the context of link prediction. Their experiments show that the simple baseline they proposed is outperforming state of the art models. They continue with the analysis of the datasets by proposing measures (features) to characterize the datasets, and the propose a non-contrastive model that outperforms other state of the art models on MRR.

**Strengths:**

1) The paper is about an interesting topic, link prediction task on temporal graph benchmark datasets.
2) The authors identify the challenges and limitation of existing models and benchmark datasets and propose measures to characterize the datasets.
3) The authors have included a fair amount of state of the art comparison methods for graph representations (link prediction task).

**Weaknesses:**

1) The paper needs more iterations to improve the writing style and the structure. It is not easy to follow.
2) The authors start the paper by introducing with the challenges and limitations of the benchmark dataset and existing models. This specific part need more attention. It is the core of the paper and it is very brief. A better listing of the limitations/challenges is needed. A related work section will help to better list those limitations. The works mentioned for the comparison (but other works too) would need to be included in the related work. This would help to motivate the importance of this analysis/study.
3) Overall, this work is not very original. I think it would be a better fit for a survey paper, where more details on the methods and the limitations will be reported and a more extensive comparison of the state of the art methods in each dataset will be included. The novelty of the proposed models is also limited.
4) The experimental analysis would be more informative if more evaluation metrics are included. The authors here include only MRR.
5) In Section 5, the sample size is very limited for the evaluation, a bigger size would be more convincing on the findings.

**Questions:**

The authors can comment in my previous listed points if they would like to.
Some more comments for the paper:
- When the term measures is first mentioned, I would expect a more emphasis on a definition and a motivation on why we need these measures and how those will help us choose a link prediction model based on the "measures" of the dataset that we would like to experiment with.
- In Table 1, most details are missing, some of them are mentioned in the text (where the table 1 is being referred to). It would be useful to add those details in the table legend as well. For example, details and descriptions for K, N, scores, coin, and comment should also be included in the Table too.
- Table 3 should be placed closer to the text that it is being referenced.

**Details Of Ethics Concerns:**

No concerns.